# Pleiotropic Effects of Influenza Vaccination

**DOI:** 10.3390/vaccines11091419

**Published:** 2023-08-25

**Authors:** Astrid Johannesson Hjelholt, Cecilia Bergh, Deepak L. Bhatt, Ole Fröbert, Mads Fuglsang Kjolby

**Affiliations:** 1Steno Diabetes Center Aarhus, Aarhus University Hospital, Palle Juul-Jensens Boulevard 11, 8200 Aarhus N, Denmark; ole.frobert@regionorebrolan.se (O.F.); mads@dandrite.au.dk (M.F.K.); 2Department of Biomedicine, Aarhus University, Høegh-Guldbergs Gade 10, 8000 Aarhus, Denmark; 3Department of Clinical Pharmacology, Aarhus University Hospital, Palle Juul-Jensens Boulevard 11, 8200 Aarhus N, Denmark; 4Clinical Epidemiology and Biostatistics, School of Medical Sciences, Örebro University, S-701 82 Örebro, Sweden; cecilia.bergh@regionorebrolan.se; 5Mount Sinai Heart, Icahn School of Medicine at Mount Sinai, One Gustave L. Levi Place, P.O. Box 1030, New York, NY 10029-6574, USA; deepak.bhatt@mountsinai.org; 6Department of Clinical Medicine, Faculty of Health, Aarhus University, Palle Juul-Jensens Boulevard 11, 8200 Aarhus N, Denmark; 7Faculty of Health, Department of Cardiology, Örebro University, SE-701 82 Örebro, Sweden

**Keywords:** influenza vaccine, pleiotropic effects, nonspecific effects, heterologous effects, cardiovascular disease, type 1 diabetes mellitus, cancer, Alzheimer’s disease, trained immunity, epigenetic modification

## Abstract

Influenza vaccines are designed to mimic natural influenza virus exposure and stimulate a long-lasting immune response to future infections. The evolving nature of the influenza virus makes vaccination an important and efficacious strategy to reduce healthcare-related complications of influenza. Several lines of evidence indicate that influenza vaccination may induce nonspecific effects, also referred to as heterologous or pleiotropic effects, that go beyond protection against infection. Different explanations are proposed, including the upregulation and downregulation of cytokines and epigenetic reprogramming in monocytes and natural killer cells, imprinting an immunological memory in the innate immune system, a phenomenon termed “trained immunity”. Also, cross-reactivity between related stimuli and bystander activation, which entails activation of B and T lymphocytes without specific recognition of antigens, may play a role. In this review, we will discuss the possible nonspecific effects of influenza vaccination in cardiovascular disease, type 1 diabetes, cancer, and Alzheimer’s disease, future research questions, and potential implications. A discussion of the potential effects on infections by other pathogens is beyond the scope of this review.

## 1. Introduction

Vaccines have saved hundreds of millions of lives and helped eradicate several diseases, especially since national programs for immunization were established and coordinated in the 1960s [1]. The current paradigm for vaccine efficacy is the induction of antigen-specific memory in adaptive immune cells, enabling long-lived protection. In addition, nonspecific effects of vaccination, also referred to as heterologous or pleiotropic effects, have opened new fields of research [2,3]. Several live vaccines, such as Bacillus Calmette-Guerin (BCG), measles and smallpox, appear to reduce all-cause mortality and hospitalization [4,5,6], and also, a number of studies imply that BCG vaccination may be protective against malignancies [7,8,9,10], allergy, and autoimmune diseases, including type 1 diabetes [11,12,13,14,15,16,17]. Likewise, non-live seasonal influenza vaccine may decrease incidence and severity of COVID-19 infection and other non-influenza respiratory infections [18,19,20,21,22,23,24,25,26,27,28].

Influenza vaccines are designed to mimic natural influenza virus exposure and stimulate long-lasting immunity. During future infections, recognition of a vaccine antigen immediately activates the innate immune system. This initial response is followed by induction of the adaptive immune system characterized by increasing titers of antigen-specific antibodies and CD4 T cells, concurrently with the upregulation and downregulation of several cytokines [29,30,31]. Different mechanisms underlying nonspecific effects of vaccines have been proposed. The adaptive immune response is suggested to contribute in two different ways: cross-reactivity, which can occur between related stimuli, such as different influenza virus strains, and bystander activation, where B and T cells are activated without specific recognition of antigens. Bystander activation can occur due to a combination of factors, including an inflammatory environment, the presence of co-signaling ligands, and interactions with nearby cells [32,33]. In addition, recent evidence has highlighted a central role for epigenetic reprogramming in monocytes and natural killer (NK) cells. These vaccination-induced epigenetic changes seem to imprint an immunological memory in the innate immune system termed “trained immunity”, inducing long-lasting effects on disease susceptibility [4,32,34].

As the influenza virus is constantly evolving, favoring mutations capable of evading host immunity, complete eradication of influenza is unlikely, and vaccination remains the most efficacious strategy to mitigate the harmful healthcare-related effects of influenza. However, the influenza vaccine may have important therapeutic potential beyond protection against influenza infection. In this review, we will discuss the possible nonspecific effects of influenza vaccination in a number of common chronic diseases. Potential effects of influenza vaccination on infections by other pathogens is beyond the scope of this review.

## 2. Literature Search

A literature search was performed in PubMed based on MeSH terms combined with free text. Search terms comprised ‘influenza vaccine’ or ‘influenza vaccination’, ‘pleiotropic effect’, ‘heterologous effect’, or ‘nonspecific effect’, in combination with cardiovascular disease, T1D, cancer, Alzheimer’s disease, or related conditions. Articles written in English were accepted. All study designs, including meta-analysis and reviews, were accepted. Manuscripts concerning the pleiotropic effects of influenza vaccination on cardiovascular disease, T1D, cancer, Alzheimer’s disease, or related conditions, as well as articles discussing the mechanisms underlying the pleiotropic vaccine-effects, were included. In addition, the references of the identified articles were searched for relevant publications, based on title.

## 3. Influenza Vaccine and Cardiovascular Disease

Cardiovascular disease, including coronary heart disease, heart failure, and cerebrovascular disease, is a major cause of mortality worldwide [35]. Atherosclerosis, which plays a central role in cardiovascular disease, is considered a chronic inflammatory disease of the vessel wall, possibly featuring an autoimmune element [36]. Although multifactorial, the inflammatory response may be modulated by exogenous pathogens [37], and evidence from numerous observational studies has identified influenza infection as a risk factor for cardiovascular disease [38,39,40,41,42,43,44,45,46]. This has led to an interest in the preventive role of influenza vaccination, and the vaccine-protective effect on cardiovascular events is well-established [47,48,49,50,51,52,53].

The most compelling data come from studies on coronary artery disease [42,49,50,51,52,53]. In addition to several observational studies, four randomized clinical trials, including a large multicenter and multinational placebo-controlled trial, named the IAMI study, involving 2571 patients over four influenza seasons [53], demonstrated that in patients with recent myocardial infarction, influenza vaccination was effective as secondary prevention, significantly reducing mortality and new cardiovascular events during one year [49,50,51,53]. In addition, influenza vaccination was safe in the acute, highly inflammatory phase after myocardial infarction [53]. Notably, a randomized clinical trial in individuals at high risk of cardiovascular disease found no additional effect on hospitalizations due to cardiac or pulmonary events or all-cause mortality when comparing the high-dose trivalent influenza vaccine with the standard-dose quadrivalent influenza vaccine [54,55], indicating no apparent dose–response relationship.

Several case–control studies have assessed the effectiveness of influenza vaccination in primary prevention of coronary artery disease by evaluating its impact on cardiovascular outcomes within the general population. These studies have consistently reported a reduced risk of hospitalization due to acute coronary syndrome in vaccinated individuals compared to non-vaccinated individuals [56,57,58]. Though prospective data addressing the effect of influenza vaccination in primary prevention of coronary artery disease are warranted, ethical limitations related to annual vaccine recommendations in older adults and certain risk groups hamper proper randomized, placebo-controlled trials. However, a randomized clinical study, currently being conducted in Denmark, compares the conventional dosage of influenza vaccine with a higher dosage (https://clinicaltrials.gov/, accessed on 10 May 2023, ClinicalTrials.gov Identifier: NCT05517174).

A similar, although somewhat weaker, association between influenza vaccination and heart failure, stroke, and transient ischemic attack in the general population has been reported [58]. While observational studies suggest that influenza vaccination may reduce mortality and hospitalization in patients with heart failure [59,60,61], a newly published large, randomized trial in heart failure patients was unable to show the effect of influenza vaccination on cardiovascular death, non-fatal myocardial infarction and stroke, and hospitalization for heart failure, whereas all-cause hospitalizations and pneumonia were significantly reduced in vaccinated as compared to unvaccinated patients [62]. No large-scale randomized, controlled trials have been performed in patients with a recent cerebrovascular event; however, observational data suggest a possible protective effect of vaccination on stroke risk [63,64,65,66,67,68].

Notably, influenza vaccination also prevents cardiovascular complications outside of influenza season [50,51,53], and the reduction in mortality risk following vaccination is greater than would be expected through the prevention of influenza infection alone, suggesting that influenza vaccination may have a nonspecific protective effect in patients with cardiovascular disease [58,69,70]. A possible explanation for this could be an immunomodulating effect of influenza vaccination, limiting dysregulated post-infarction inflammation [70]. In a smaller randomized, placebo-controlled trial, influenza vaccination one week before a coronary artery bypass grafting operation, caused a substantial reduction in proinflammatory cytokine levels (IL-6, IL-8 and TNF-α), whereas the anti-inflammatory marker IL-10 was five-fold increased in vaccinated patients, attenuating the inflammatory response related to the procedure [71]. Interestingly, data indicate that patients with a recent cardiovascular event have a higher treatment effect of vaccination [72], and vaccine effectiveness appeared higher in non-ST-segment myocardial infarction, which is associated with increased inflammation and macrophage activation, as compared with ST-segment myocardial infarction [73,74]. Another explanation for the pleiotropic effects of influenza vaccination in protection of cardiovascular events is cross-reactivity [70]. Possible candidates for cross-reactivity are the bradykinin B2 receptor, which are known to be important in maintaining cardiovascular homeostasis [75], and apolipoprotein B, a surface protein on low-density lipoprotein (LDL) cholesterol particles, which are highly correlated with clinical atherosclerosis [37,70].

Further understanding of the mechanisms involved in the protective effect of vaccination patients with cardiovascular disease would be highly important and valuable.

## 4. Influenza Vaccine and Type 1 Diabetes

Type 1 diabetes mellitus (T1D) is a chronic autoimmune disease in which autoreactive T cells attack insulin-producing pancreatic β-cells. This leads to a loss of β-cell function, and as the disease progresses, exogenous insulin is necessary to control blood glucose [76]. Although ongoing advancements in insulin therapy have improved management of T1D, insulin therapy does not fully prevent complications, and patients with T1D have an increased risk of cardiovascular disease [77], foot ulcers and amputations, diabetic retinopathy [78], and kidney failure [79]. Importantly, children with residual β-cell function experience lower risk of severe hypoglycemia, have better blood glucose control, and lower insulin requirements [80], leading to better diabetes management. Therefore, in addition to insulin treatment, disease-modifying therapies are needed.

Type 1 diabetes is preceded by a preclinical phase, during which β-cell function is sufficient to maintain glycemic control. Identifying and treating individuals at this preclinical stage potentially provides a therapeutic window to slow down the progression of β-cell destruction, delaying the need for insulin therapy [81]. In theory, immunotherapy aiming to modulate the immune system and avoid β-cell destruction is a promising strategy to prevent or delay the onset of T1D. Most potentially modulating drugs have been repurposed from the transplantation field or other autoimmune diseases but the success rate of immunotherapy treatment in T1D has been low. While T1D can be well regulated with insulin treatment, any disease modifying therapy should be safe. Therefore, chronic immunosuppression regimens, which often have adverse effects related to host immune defense and tumorgenicity, are deemed unacceptable [76,81,82].

In individuals genetically predisposed to T1D, environmental factors, such as a viral infection, may trigger autoimmunity, and several lines of evidence suggest that infection with influenza A (H1N1) virus is associated with the onset of T1D [83,84,85,86,87]. Interestingly, a prospective observational study conducted in two countries investigating the risk of β-cell autoimmunity in genetically disposed children following exposure to the inactivated influenza vaccine Pandemrix^®^, reported a lower risk of T1D in vaccinated compared to non-vaccinated children in Finland, whereas no difference was observed in Sweden, suggesting that the vaccine may reduce the risk of T1D in selected populations [88]. This possibly reflects the prevention of triggering viral infections in influenza-vaccinated children, but divergent results have been reported concerning the effectiveness of influenza vaccination in patients with T1D [89,90], emphasizing the need for randomized controlled trials [91]. It may also be speculated that the reduced risk of islet autoimmunity in influenza-vaccinated children could be explained by pleiotropic effects. Influenza vaccination may help redirect the immune system by inducting a strong immune response towards the vaccine, thereby alleviating the immunologic reaction against β cells. This line of thinking has been successfully implemented in cancer therapy [92] and is in accordance with new insights favoring engagement rather than suppression of the immune system to reverse the immunopathogenesis of T1D [93]. In a recent authoritative review discussing immunotherapy in T1D, the authors proposed targeting of cytokine antagonists in the prevention of islet autoimmunity, including interleukin (IL)-6, IL-8 and tumor necrosis factor-α (TNF-α) and agonist targets, such as IL-2 and IL-10 [76]. Following influenza vaccination, several cytokines are upregulated or downregulated. In healthy volunteers, a three-fold rise in IL-2 is observed 6 weeks following immunization [94], while IL-6 and TNF-α are practically unchanged one week after vaccination [95]. During inflammation, influenza vaccination leads to reduced levels of proinflammatory cytokine (IL-6, IL-8 and TNF-α) and an increase in IL-10 levels [71]. Thus, the immune response to influenza vaccination affects five putative T1D cytokine targets in the right direction [76]. The findings of a recent study analyzing the epigenomic and transcriptional consequences of influenza immunization in humans indicate the presence of epigenomic changes, potentially modulating long-term cytokine profile. The implications of this discovery for the possibility of influencing the course of T1D may be significant [96].

Inspired by these findings, we are conducting a clinical trial in children with new onset T1D to evaluate the effect of influenza vaccination in sustaining β-cell function (NCT05585983).

## 5. Influenza Vaccine and Cancer

The treatment of cancer was revolutionized by the introduction of immunotherapy, which is still the subject of intense research. The microenvironment of the tumor is important for the efficacy of currently available immunotherapies, including immune checkpoint inhibitors [97]. An immunologically “hot” tumor microenvironment, characterized by accumulation of proinflammatory cytokines and T cell infiltration, exhibits a strong response rate to treatment with immune checkpoint inhibitors, as opposed to a “cold” tumor microenvironment. Several strategies are currently being investigated to promote the transformation of “cold” tumors into “hot” tumors [92].

Influenza infection is associated with a higher risk of morbidity and mortality in patients with cancer [98], and observational studies suggest that annual influenza vaccination reduces incidence of lung cancer [99,100]. A recent register-based cohort study found that in patients undergoing curative surgery for solid tumors, postoperative influenza vaccination is associated with reduced overall and cancer-related mortality [101]. Despite safety concerns, large-scale observational studies conclude that influenza vaccination in combination with immunotherapy is safe, and data indicate that the vaccine may improve overall survival in patients treated with immune checkpoint inhibitors [98,102,103,104,105,106].

Mechanistically, influenza vaccination is suggested to modulate anti-tumor immune responses, thereby reducing tumor growth. Preclinical evidence indicates that intratumoral injection of the unadjuvanted influenza vaccine limits tumor progression in mice, sensitizes tumors resistant to checkpoint blockade, and increases the proportion of tumor antigen-specific CD8+ T cells and dendritic cells within the microenvironment of the tumor. This suggests that immunologically “cold” tumors may be converted to “hot” tumors [92]. The perioperative use of influenza vaccine is also supported by preclinical studies. Following cancer surgery, NK cell cytotoxicity, which plays a crucial role in tumor clearance, is impaired, and this surgery-induced NK cell dysfunction is associated with a higher rate of cancer recurrence and mortality [107]. In murine models, preoperative influenza vaccination prevented NK cell dysfunction and reduced the metastatic dissemination of cancer cells [108]. Notably, the reduction in metastasis and the ex vivo NK cell cytotoxicity was greater when perioperative influenza vaccination was administrated in combination with the phosphodiesterase-5 inhibitor sildenafil as compared to either treatment alone [109]. This may be interesting in the context of other pleiotropic uses of the influenza vaccine and could be an interesting subject for future research. Preoperative influenza vaccination was accompanied by an increase in IFN-α, IL-2, IL-12, and IL-15, and additional experiments in type I interferon receptor knockout mice and human peripheral blood mononuclear cells (PBMCs) suggest that the influenza vaccine effect on NK cell activity may act through the modulation of IFN-α [108]. A growing interest in the anti-tumor properties of cytokines has led to numerous clinical trials evaluating cytokine-based drugs, and IFN-α and IL-2 have been approved in the treatment of several malignancies [110].

Influenza vaccination seems a promising approach in oncologic therapy, and clinical trials evaluating safety and efficacy of an intratumoral influenza vaccine before intended curative surgery (NCT04591379) and influenza vaccination on the day of surgery in combination with the phosphodiesterase-5 inhibitor tadalafil [109] (NCT02998736) in patients with abdominal cancer are ongoing.

## 6. Influenza Vaccine and Alzheimer’s Disease

Alzheimer’s disease, the most common type of dementia, represents a progressive neurodegenerative condition characterized by an initial decline in short-term memory, leading to significant cognitive impairments caused by extensive neuronal dysfunction. Extracellular accumulation of amyloid-beta plaques and intracellular neurofibrillary tangles, which consist of abnormally hyperphosphorylated tau protein together with chronic low-grade inflammation, are hallmarks of Alzheimer’s disease [111]. Neuroinflammation may be crucial in the pathogenesis of Alzheimer’s disease, and microglia, which are resident innate immune cells of the brain, are considered intricately involved in the pathogenesis [111]. While the contribution of microglia in amyloid-beta clearance is beneficial, the concomitant activation and subsequent release of proinflammatory cytokines, including IL-1b, IL-8 and TNF-α results in sustained low-grade inflammation, which is an essential feature of various neurodegenerative disorders, including Alzheimer’s disease [111,112].

Evidence from a number of smaller observational studies, including a meta-analysis, suggests that influenza vaccination may be associated with a lower risk of dementia [113,114,115,116]. This was supported by a large cohort study involving 1,185,611 vaccinated and 1,170,868 unvaccinated patients using propensity score matching, reporting a 40% reduced risk of Alzheimer’s disease among influenza-vaccinated elderly persons during the 4-year follow-up period [117]. Data from experiments in mice have shown that changes in peripheral cytokine secretion associated with influenza vaccination directly affect microglial activity and amyloid-beta clearance in the *APP*/*PS1* mouse model of Alzheimer’s disease [118]. In line with this, several other vaccines, including tetanus and diphtheria [116,119], zoster [120], and BCG [121,122], have also been associated with a reduced risk of dementia, suggesting that the protective impact of influenza vaccination on Alzheimer’s disease is due to nonspecific effects on the immune system rather than direct antigen-specific protection against infection. Long-term prospective studies and randomized controlled trials, however, are missing.

## 7. Discussion

Several lines of data suggest that the use of the influenza vaccine as a safe and cost-effective drug repurposing strategy may be protective in the development and/or progression of a variety of chronic diseases (See Table 1 for a comprehensive overview). Findings from vaccine safety monitoring systems consistently demonstrate that influenza vaccines exhibit a remarkable level of safety. Typically, any side effects experienced are mild in nature and encompass temporary discomfort, such as pain, redness, and swelling at the injection site, as well as symptoms like headache, fever, muscle aches, joint pain, or fatigue [123]. Only a few studies have been conducted to explain the possible molecular mechanism underlying the nonspecific effects of influenza vaccination. In cardiovascular disease, T1D, and Alzheimer’s disease, chronic low-grade inflammation is essential in disease pathogenesis [70,76,111,112], and the potential beneficial effects of influenza vaccine may be attributed to immunomodulation of the innate immune system, including up regulation and downregulation of several cytokines, possibly in combination with cross-reactivity and bystander activation of the adaptive immune system [21,33,75,96,124,125]. In contrast, in cancer, vaccination seems to be immune activating, creating an immunologically inflamed hot microenvironment and increasing the antitumor immune response [92].

Vaccine-induced epigenetic modifications in monocytes and NK cells, leading to modulation of gene expression and long-lasting trained immunity in the innate immune system, are most extensively studied in association with the BCG vaccine [4,32,34]. However, recent evidence points to an important role of trained immunity following influenza vaccination as well [21,96]. In a recent study, influenza vaccination in human volunteers was associated with transcriptional changes in lymphoid and myeloid cells together with downregulation of inflammatory biomarkers as well as reduced pro-inflammatory and increased anti-inflammatory cytokine production [21]. In support of an influenza vaccine-induced epigenomic remodeling of the innate immune system, Wimmers et al. reported decreased histone H3 Lys27 acetylation (H3K27ac) in monocytes and myeloid dendritic cells and reduced cytokine responses to Toll-like receptor (TLR) stimulation following influenza vaccination [96]. Interestingly, vaccination with the AS03-adjuvanted influenza vaccine induced a similar response, but in addition, the vaccine increased chromatin accessibility in monocytes and myeloid dendritic cells, which was associated with elevated expression of antiviral genes and improved resistance to various viruses [96]. These findings suggest that different types of vaccines and use of adjuvants may elicit unique trained immunity programs. As modulation of cytokine secretion seems pivotal, a possible disease-modifying effect of influenza vaccination could also play a role in other conditions characterized by chronic low-grade inflammation or autoimmunity, such as type 2 diabetes and migraine, in which elevation of cytokines, including TNF-α, IL-6 and IL-10, are associated with disease progression [124,125].

Findings by Veljkovic et al. suggest that influenza vaccination promotes nonspecific cross-reactivity between antibodies and the human bradykinin B2 receptor [75]. The interaction stimulated the production of nitroxide (NO), which increased myocardial perfusion and induced cardio-protection, offering an explanation for the preventive effect of influenza vaccination in cardiovascular disease. Bystander activation, characterized by antigen-independent activation of B and T cells, may also be implicated in influenza-vaccine pleiotropy. A study of influenza-vaccinated individuals demonstrated several antibody clones from purified B cells that did not bind vaccine antigens, suggesting that vaccination induced activation of antibody clones in an antigen-independent manner, indicative of bystander activation of memory B cells [33]. Despite these promising results, several questions remain unanswered. First, the timing as well as the frequency of vaccination are important considerations. Recent evidence suggests that influenza vaccination has the potential to influence the signaling of cytokines in response to different stimuli, and it may be speculated that inflammation influences vaccine-response [73]. Also, knowledge about effect durability is pivotal. Previous studies report reduced mortality and incidence of new cardiovascular events up to 12 months following vaccination [53], and indications of epigenetic modifications can be detected for up to 6 months [96]; however, whether the effects persist remains unknown, necessitating randomized trials with a long follow-up time. Second, existing research indicates that different types of influenza vaccines, e.g., adjuvanted vaccines, induce distinct immune responses [96,126], highlighting the complexity of vaccine-induced epigenomic reprogramming of immune cells. In this context, “immunosenescence”, referring to an age-related decline in immune function, is worth mentioning. Clinical data indicate that influenza vaccination is less protective against influenza infection in older adults as compared to working-age adults [127], and for individuals older than 65 years, high-dose and adjuvanted vaccines are available [128]. Interestingly, in the IAMI study, the effect of the influenza vaccine on all-cause death and new cardiovascular events was higher in patients older than 65 years as compared to patients younger than 65 years, although this did not reach statistical significance [53]. Likewise, in a randomized trial in patients at high risk of cardiovascular disease with a median age of 65.5 years, no difference in vaccine-effect on hospitalizations and all-cause mortality was demonstrated when comparing high-dose with standard-dose influenza vaccination. This suggests that immunosenescence may not translate into a reduction in the nonspecific effects of influenza vaccination [55]. Whether the same is evident with the adjuvanted vaccine is unknown, and future research should take these considerations into account.

Finally, although data from randomized trials in coronary artery disease are very compelling, most of the existing evidence on the positive nonspecific vaccine effects is derived from observational data. It is important to acknowledge the risk of potential bias, including the risk of ‘healthy user bias’, and results from observational studies must be interpreted with caution. To obtain a full picture of the possible pleiotropy of influenza vaccination, additional studies focusing on the immunological and molecular mechanisms underlying the effects, including timing and frequency of vaccination, as well as the influence of adjuvants, are needed. To this end, more randomized controlled trials are essential.

## Figures and Tables

**Table 1 vaccines-11-01419-t001:** Effects of influenza vaccination in primary and secondary prevention of coronary artery disease, heart failure, type 1 diabetes, cancer, and Alzheimer’s disease in humans. Randomized, controlled trials are listed. In absence of randomized trials, observational studies are listed.

Disease	Effect of Influenza Vaccination	Randomized Controlled Trials	Observational Studies
**Primary prevention of coronary artery disease**	Reduced risk of hospitalization for acute coronary syndrome		Puig-Barbera et al. [56] Siriwardena et al. [57] Davidson et al. [58]
**Secondary prevention of coronary artery disease**	Reduced mortality and cardiovascular events	Phrommintikul et al. [49] Ciszewski et al. [50] Gurfinkel et al. [51] Frobert et al. [53]	
**Heart failure**	Reduced all-cause hospitalizations and pneumonia	Loeb et al. [62]	
**Stroke**	Reduced stroke risk		Holodinsky et al. [63] Nichol et al. [64] Lin et al. [65] Lavallee et al. [66] Asghar et al. [67] Lam et al. [68]
**Type 1 diabetes**	Reduced disease risk		Elding Larsson et al. [88]
**Cancer**	Reduced risk of lung cancer Reduced overall and cancer-related mortality in patients undergoing curative surgery Increased overall survival (in combination with immune checkpoint inhibitors)		Chen et al. [99] Chen et al. [100] Gogenur et al. [101] Bersanelli et al. [106]
**Alzheimer’s disease**	Reduced disease risk		Liu et al. [114] Luo et al. [115] Verreault et al. [116] Bukhbinder et al. [117]

## Data Availability

No new data were created or analyzed in this study. Data sharing is not applicable to this article.

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
