# Peer review of "Pleiotropic Effects of Influenza Vaccination"

_vaccines, 2023, doi:10.3390/vaccines11091419_

Round 1

Reviewer 1 Report

Here, we are presented with a review paper covering influenza vaccination's potential positive side effects. As the authors state, there is no clear-cut explanation or known biological process for the apparent side effects. My first thoughts go in the direction of the “hygiene hypothesis”. In this case, the vaccine skews the immune system toward responding toward the vaccine and away from less desired inflammatory activity in the body. 

Do you think vaccination is viable for treating the various medical conditions mentioned in the review?

I would be sceptical of such an approach since, firstly, there would be questionable how long the effect will last, and secondly, the risk of adverse effects.

I do not have any further comments.

none

Reviewer 2 Report

This is a review about non specific effects of possible pleiotropic effects of influenza vaccines as a prevention not only against influenza infection, but as a repurposing preventive stratery even for non infectious autoimmune and other diseases. The most well studied relationship is  the effect of the vaccine with the cardiovascular events, without well known mechanisms.

Recently BCG vaccine has been studied as a preventive strategy for COVID-19 infection, before the availability of specific vaccines, based on preclinical data in murine models, with contradictory results, a negative impact on the incidence on the incidence of COVID-19 infection, but protective effect on influenza incidence and other viral infections.

In several studies, although the protection from severe respiratory infection is more crucial in the elderly population and immunocompromised patients, there is an age-related decline in immune response, known as immunosenescence. Therefore, in order to study any benefit of the trained immunity you should adjust for many variables, such as age, immunocompromised status, concurrent therapies, such as anti-TNF regimens, which could diminish any effect.

Apart for the data of the benefit of the vaccination on CVD risk, the association with autoimmune disease, dementia, cancer is very weak, compared even with the data of over the counter drugs, such as probiotics and should be emphasized in the manuscript and you should be focused on future basic research that should be done.

The present review is only a trigger factor for further studies, although is unethical to conduct placebo control trials in the context of hesitancy of life-saving vaccines, especially in populations at risk.

Reviewer 3 Report

The manuscript by Hjelholt et al. is a comprehensive review of the past and recent findings on the pleiotropic effects of the influenza vaccine, i.e., how the influenza vaccine induces bystander immune alterations, for example a change in the pattern of NK cell or innate immune cell responses. Overall, the review is well-written, and provides a complete overview of these findings.

- The only minor comment I have is a suggestion: authors could make and provide a table summarizing the effects of influenza vaccine in different chronic diseases listed. That table could also include the clinical trials listed in the text.

Otherwise, I'd recommend acceptance of the manuscript.

Reviewer 4 Report

Interesting review on pleiotropic/non-specific effects of influenza vaccines on certain chronic diseases: cardiovascular diseases, type 1 diabetes, Alzheimer's and cancer. Results are gained from observational studies. Explanations of possible mechanisms of action by immune modulation are given. Outlook on more data needed like timing, frequency and type of influenza vaccine is given. References on other vaccines, mainly BCG, which have known pleiotropic effects, are mentioned.

Reviewer 5 Report

A timely review of an important topic. It is very well written.  I noticed that the authors do not mention effects of influenza vaccination on infections by other pathogens (which became particularly relevant during the SARS-CoV-2 pandemic). If the authors are of the opinion that it would go beyond the scope of this review to discuss this topic, it would be helpful if they stated clearly in Abstract and Introduction that they limited the review to effects on non-transmissable disease entities. Also, there should be a brief section titled Methods which describes the search strategy used to include / exclude studies for this review. Regarding line 162, it would be better to write "may reduce the risk of T1D in selected populations"

Round 2

Reviewer 2 Report

In the revised manuscript have been included several clarifications and therefore should be considered as appropriate for publication